# Effect of Radiation Defects on Thermo–Mechanical Properties of UO$_2$ Investigated by Molecular Dynamics Method

**Ziqiang Wang [1,2], Miaosen Yu [2], Chen Yang [1], Xuehao Long [2], Ning Gao [2,3,*], Zhongwen Yao [4,*], Limin Dong [5] and Xuelin Wang [2]**

[1]  Key Laboratory of Bionic Engineering, Ministry of Education, Jilin University, Changchun 130022, China; wangziqiang2021@gmail.com (Z.W.); chenyangjlu@126.com (C.Y.)

[2]  Institute of Frontier and Interdisciplinary Science, Key Laboratory of Particle Physics and Particle Irradiation (MOE), Shandong University, Qingdao 266237, China; 202120966@mail.sdu.edu.cn (M.Y.); 17862969617@163.com (X.L.); xuelinwang@sdu.edu.cn (X.W.)

[3]  Institute of Modern Physics, Chinese Academy of Sciences, Lanzhou 730000, China

[4]  Department of Mechanical and Materials Engineering, Queen's University, Kingston, ON K7L3N6, Canada

[5]  School of Materials Science and Chemical Engineering, Harbin University of Science and Technology, Harbin 150080, China; donglimin@hrbust.edu.cn

*  Correspondence: ning.gao@sdu.edu.cn (N.G.); zwyao@jlu.edu.cn (Z.Y.)

**Abstract:** Nuclear fuel performance is deteriorated due to radiation defects. Therefore, to investigate the effect of irradiation-induced defects on nuclear fuel properties is essential. In this work, the influence of radiation defects on the thermo-mechanical properties of UO$_2$ within 600–1500 K has been studied using the molecular dynamics method. Two types of point defects have been investigated in the present work: Frenkel pairs and antisites with concentrations of 0 to 5%. The results indicate that these point defects reduce the thermal expansion coefficient ($\alpha$) at all studied temperatures. The elastic modulus at finite temperatures decreases linearly with the increase in concentration of Frenkel defects and antisites. The extent of reduction ($R$) in elastic modulus due to two different defects follows the trend $R_\text{f} > R_\text{a}$ for all studied defect concentrations. All these results indicate that Frenkel pairs and antisite defects could degrade the performance of UO$_2$ and should be seriously considered for estimation of radiation damage in nuclear fuels used in nuclear reactors.

**Keywords:** uranium dioxide; Frenkel pairs; antisites; molecular dynamics; elastic modulus; thermal expansion

## 1. Introduction

Uranium dioxide (UO$_2$) is widely used as a nuclear fuel in the nuclear industry for various nuclear power reactors [1]. Thus, the safe operation of a nuclear reactor correlates strongly with the stability of UO$_2$. However, under extreme conditions, different radiation defects (e.g., vacancies, interstitials, and voids) would be created within the nuclear fuel due to irradiation. These defects would lead to severe degradation of the physical, thermal, and mechanical properties of the nuclear fuels [2–4]. For example, irradiation-induced fission products and vacancies can produce bubbles and voids, causing swelling and fragmentation which thus deteriorates the performance of fuels [5]. Therefore, to investigate the effect of radiation-induced defects on the thermo-mechanical properties of uranium dioxide is essential.

In the literature, numerous experimental and theoretical studies have been performed to understand the impact of fission products, porosities, and other defects on thermal transport in UO$_2$. For example, Hobson et al. analyzed porous UO$_2$ with porosity levels of 4.11 to 8.58% and observed the relationship of reductions in thermal conductivity to the temperature [6]. An experimental study on the effect of soluble fission products on thermal conductivity was also performed, which found that at lower temperatures the thermal

conductivity would decrease with an increase in fission product concentration; however, at higher temperatures the concentration of fission products has only a slight influence on thermal transport [7].

In addition to experiments, computer simulations have also provided an effective tool to analyze the specific mechanism of fuels at the atomic level. Liu et al. investigated the effect of uranium vacancies, oxygen vacancies, and fission products on the thermal conductivity of uranium. They also observed a stronger effect on the reduction in thermal conductivity by uranium vacancies compared to that by oxygen vacancies [8]. Chen et al. performed molecular dynamics (MD) simulations to examine the effect of Xe bubble size and pressure on the thermal conductivity of uranium dioxide and demonstrated that the dispersed Xe atoms could result in a lower thermal conductivity than by clustering them into bubbles [9]. Uchida et al. performed molecular dynamics simulations to evaluate the effect of Schottky defects on the thermal properties in $UO_2$ and reported that thermal conductivity decreased with the increasing concentration of Schottky defects [10]. Furthermore, the thermal transport of $ThO_2$, as an alternative to conventional uranium nuclear fuel, was also investigated extensively. For example, Park et al. [11] investigated the effect of vacancies and substitutional defects on the thermal transport of $ThO_2$ by employing reverse non-equilibrium molecular dynamics (NEMD). The authors reported that the effect of thorium vacancy defects on the thermal transport of $ThO_2$ is even more detrimental than that of oxygen vacancy defects. In addition, compared to vacancy defects, substitutional defects in $ThO_2$ slightly affect the thermal transport [11]. To investigate the effect of irradiation-induced fission products on the thermal conductivity of thorium dioxide, Rahman et al. [5] examined the effect of Xe and Kr with impurity concentrations of 0 to 5% on the thermal conductivity of $ThO_2$ with the molecular dynamics method, and found that Xe and Kr resulted in significant reductions in the thermal conductivity of $ThO_2$.

In order to use nuclear fuels safely in reactors, the mechanical feature of fuels after irradiation is also an important property which needs to be considered [12]. For example, Jelea et al. examined the thermo-mechanical properties of a $UO_2$ matrix containing different concentrations of porosity and observed that the elastic modulus decreased with an increase in porosity concentration [13]. Rahman et al. examined the effect of fission products (Xe and Kr) and porosity on mechanical properties of $ThO_2$ within 300–1500 K using molecular dynamics simulations. By comparing the effect of fission products and porosity, the authors reported that the fission products resulted in a stronger reduction in elastic modulus than the porosity [14].

Although the effects of fission products and porosity on thermo-mechanical properties of $UO_2$ have been studied by different groups, to our best knowledge no investigation has been performed about the effect of Frenkel defects and antisites on the thermo-mechanical properties of irradiated $UO_2$. Considering its importance, in this work, the influences of Frenkel defects and antisites on the thermal expansion coefficient and elastic modulus of uranium dioxide are investigated extensively via molecular dynamics simulations. The thermal expansion coefficient of perfect and damaged systems is evaluated from changes in lattice parameters. Three independent elastic constants are calculated for each system, which are used to estimate the elastic modulus. The reduction in the elastic modulus induced by Frenkel defects and antisites is also calculated as a function of concentrations of defects in the system. A comparison is finally made between the effects of Frenkel defects and antisite defects to provide more understanding about the structure and property changes of $UO_2$ after irradiation. In the following sections, the computational method is first presented. The results and discussion are provided in Section 3. The conclusion is made in the last section.

## 2. Computational Method

In this work, MD simulations are performed using the LAMMPS (Large scale Atomic/Molecular Massively Parallel Simulator) code (Sandia National Laboratories, Livermore, CA, USA) [15], which is a classical molecular dynamics (MD) code used to simulate the

atomic interaction of selected materials. The interatomic potential developed by Cooper, Rushton, and Grimes (CRG) is used in this work for U-U, U-O and O-O interactions [4], which has been proven to reliably predict the various thermo-mechanical properties within the temperature range of 300 K to 3000 K [8,16–18]. In order to accurately describe the properties of $UO_2$, the Coulomb interaction is further included with the original pair. The computational box used in this work is a $10 \times 10 \times 12$ extension of fluorite ($CaF_2$) unit cells containing 14,400 atoms. The lattice parameter for the computational box is the equilibrated lattice parameter at the investigated temperature. The periodic boundary condition is applied in all directions.

Generally, the O/U ratio in all defects after a displacement cascade in $UO_2$ is close to two [19], which is in agreement with the results presented by Devanathan et al. [20] and Van Brutzel et al. [21]. In order to create this structure of uranium dioxide with Frenkel defects and maintain the neutral charge of the system, uranium and oxygen atoms are removed from the system by keeping 1:2 ratio. The same amount of uranium and oxygen atoms are then randomly inserted at the octahedral interstitial positions of the face-centered cubic (fcc) cation sublattice [22]. Different from Frenkel defects, oxygen-antisites are created by substituting O atoms with U atoms. Similarly, uranium-antisites are created by substituting U atoms with O atoms. In order to maintain the stoichiometry of the defected system, the number of O-antisites is equal to that of U-antisites. In order to investigate the effect of defect concentration, $UO_2$ structures with 1%, 2%, and 5% Frenkel defects and 1%, 2%, and 5% antisite defects are built for further simulations. It should be noted that in the present work the concentration is defined as the value before MD relaxation at given temperatures. The main reason is that the relaxations at different temperatures could result in different concentration values after full relaxation. In order to avoid this misunderstanding during the investigation of concentration effects in this work, the concentration value before MD relaxation is used. For each defect concentration the statistical results are made based on 3 samples by randomly creating Frenkel or antisite defects.

The simulation box is first relaxed by the conjugated gradient (CG) method at 0 K and further relaxed for 500 ps at the temperature of interest under NPT (constant number, pressure, and temperature) ensemble with zero external pressure. It should also be noted the system containing 5% defects requires longer equilibration time (600 ps) to reach the equilibrium state. The equilibration is checked by the dependence of total energy and volume of the system on simulation time, which both indicate that the system has reached the equilibrium state before further calculations of lattice constants, thermal expansion coefficient, and elastic modulus. Therefore, the results obtained in the present work are reliable and could provide useful information to understand the properties of $UO_2$ after irradiation. The timestep of 1 fs is used for all simulation processes.

The lattice parameter as a function of temperature for different concentrations of Frenkel defects and antisite defects is first calculated. The thermal expansion coefficient ($\alpha$) is then determined from the first derivate of the lattice parameter with respect to the temperature using the following Equation (1):

$$\alpha(\text{T}) = \frac{1}{L}\left(\frac{\partial L}{\partial T}\right)_p \tag{1}$$

where $L$ is lattice parameter and $\left(\frac{\partial L}{\partial T}\right)_p$ is the slope of the plot for the lattice parameter as a function of temperature at the given temperature [23]. It should be noted that, in the present work, the structure of $UO_2$ is FCC and thus the thermal expansion coefficient is isotropic.

As the structure of uranium dioxide is cubic, three independent elastic constants ($C_{11}$, $C_{12}$, and $C_{44}$) need to be calculated. These elastic constants can be calculated by applying elementary strain in six directions and measuring the changes in the six stress components. In this work, the strain to induce the deformation of the simulation box was set to be $10^{-5}$. Based on the dependence of the stress on the strain, these constants are calculated as described in [24]. Based on these three constants, the bulk modulus ($B$), shear modulus

(G), and Young's modulus (Y) can be calculated. The bulk modulus is calculated with the following equations [14].

$$B = (C_{11} + 2C_{12})/3 \qquad (2)$$

Furthermore, according to the Hill's suggestion [25], in order to determine the shear modulus, the shear modulus ($G_V$) using the Voigt method and the shear modulus ($G_R$) using the Reuss method need to be obtained, which can be determined using the following Equations (3) and (4), respectively.

$$G_V = (C_{11} - C_{12} + 3C_{44})/5 \qquad (3)$$

$$G_R = (5(C_{11} - C_{12})C_{44})/(4C_{44} + 3(C_{11} - C_{12})) \qquad (4)$$

Thus, the shear modulus (G) using Hill's method can be obtained as the arithmetic average of $G_V$ and $G_R$.

$$G = (G_V + G_R)/2 \qquad (5)$$

Based on the calculated bulk modulus and shear modulus, Young's modulus is calculated with the following Equation (6).

$$Y = 9BG/(3B + G) \qquad (6)$$

## 3. Results and Discussion

### 3.1. Effect of Defects on Lattice Parameter and Thermal Expansion Coefficient

The lattice parameter (L) of pure $UO_2$ as the function of temperature is plotted in Figure 1. The error bars in the figure correspond to the standard deviation calculated among the five different statistical lattice constants calculated at the given temperature. For comparison, the results from the VASP calculation by Wang et al. [26] and from the experimental measurement by Taylor et al. [27], Yamashita et al. [28], and Momin et al. [29] are also included in Figure 1. It is also clear that L increases linearly with increases in temperature from 0 K to 1500 K as investigated in this work. From these results, the results from the present MD agree better with the experimental value than those from VASP calculations. Thus, the MD method and the related empirical potential could be used for the present purpose for further simulations.

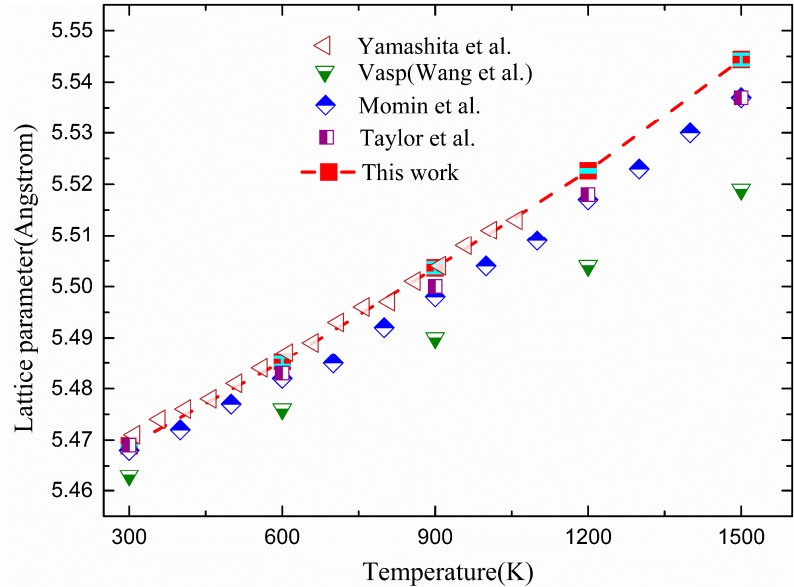

**Figure 1.** Dependence of lattice constant on temperature including the results from the present work, the experiment studies by Taylor et al. [27], Momin et al. [29], Yamashita et al. [28], and the VASP results by Wang et al. [26].

When defects are formed after irradiation, the effects of radiation defects are also simulated in this work. In Figure 2, the dependence of the lattice parameter of $UO_2$ on defect concentration at different temperatures is provided for Frenkel defects (dash) and antisite defects (solid). From Figure 2, it is clear that the lattice parameter of the system increases with an increase in Frenkel defect concentration from 0 to 5% at all investigated temperatures in the present work, although the increases are limited around 1.0%.

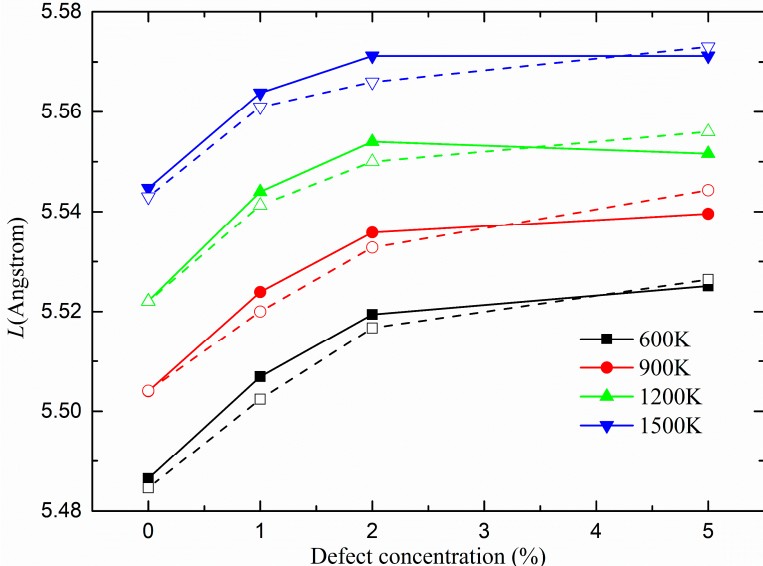

**Figure 2.** Dependence of lattice constant of $UO_2$ on defect concentration at different temperatures for Frenkel defects (dash) and antisite defects (solid).

The present results indicate that the formation of Frenkel pairs or antisite defects could increase the lattice constant quickly when the defect concentration is less than 2.0%, above which the lattice constant is almost a constant or increases or decreases slightly. Thus, the results are different from previous studies about the effect of porosity on the lattice parameter $L$ of $ThO_2$, which reported that the $L$ of the system increased linearly with increases in porosity concentration. The reason for the above difference may be mainly from the property of anisotropic effects induced by antisite defects and interstitials in Frenkel pairs, which is different from the isotropic vacancy or porosity. It should also be noted here that in this work, only defect concentrations up to 5% are considered. If higher concentrations were considered, the lattice constant may change accordingly.

As stated previously, because of the high temperature within the fuel due to the fission reaction, thermal expansion coefficient is considered to be an important factor for modelling and predicting the nuclear fuel's behavior [13]. Figure 3 presents the effects of Frenkel defects and antisite defects on the thermal expansion coefficient of $UO_2$ as a function of temperature. The uncertainty of the thermal expansion coefficient at different temperatures has also been calculated by changing the defect distribution but keeping the same concentration as stated in the computational method section. As shown in Figure 3, the uncertainty is limited for the three cases investigated in the present work. For comparison, the experimental results [27], MD derived values [30], and first principles data [31] for pure $UO_2$ are included in the figure.

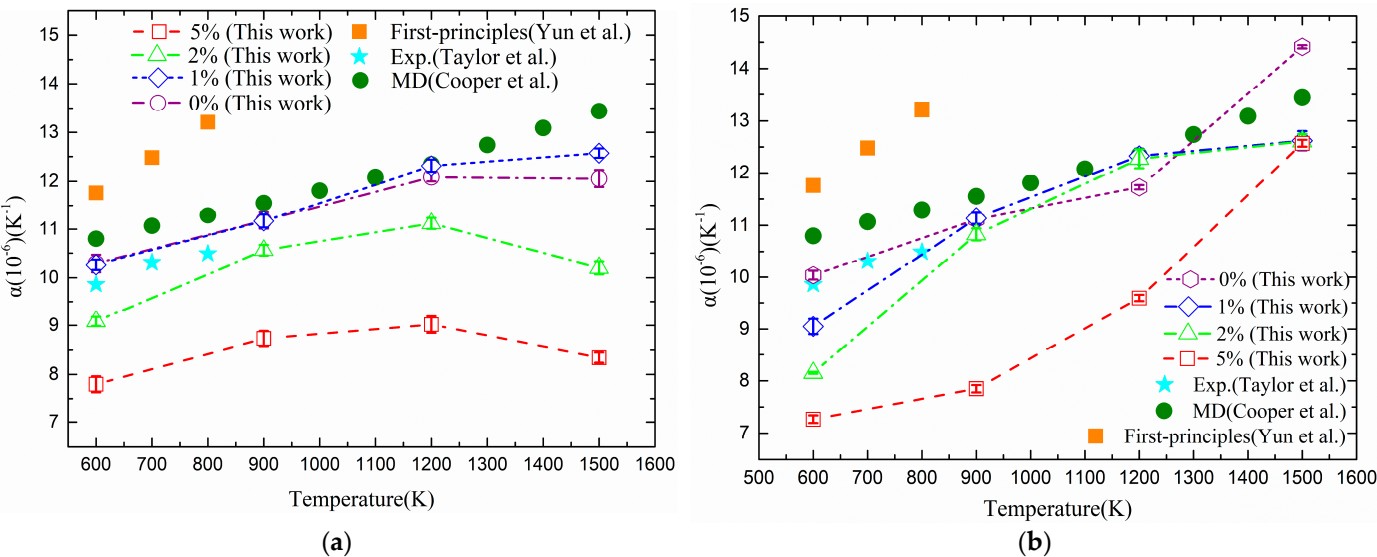

**Figure 3.** Thermal expansion coefficient for pure $UO_2$ and for defective $UO_2$ with different percentages of Frenkel defect (**a**), and antisites (**b**), as a function of temperature. For comparison, the experiment studies by Taylor et al. [27], the MD results by Cooper et al. [30] and the first-principles data by Yun et al. [31] are included in the figure.

Figure 3a shows that compared to the perfect system, the thermal expansion coefficient of systems containing 1% Frenkel defects has similar results to that of perfect systems at temperatures from 600 K to 1200 K, while at 1500 K the 1% Frenkel defects induce larger thermal expansion coefficients with an increase around 4.0%. When the system contains a higher concentration of Frenkel defects, e.g., 2.0% and 5.0%, the lower thermal expansion coefficients are observed. For example, when the concentration of Frenkel defects is 2%, the maximum reduction of thermal expansion coefficient up to 15% is observed at 1500 K. When Frenkel defect concentration is 5%, the maximum reduction is around 30% within 600–1500 K. Thus, it is found that the reduction of the thermal expansion coefficient increases with the increase in the concentration of Frenkel defects. Furthermore, it can also be observed from Figure 3a that when the defect concentration is low, e.g., 1%, the thermal expansion coefficient increases with an increase in temperature, while the higher concentration defects (2% and 5%) result in an increase in the thermal expansion coefficient for systems from 600 K to 1200 K, but decrease from 1200 K to 1500 K. In fact, according to Sun et al. [32], the thermal expansion coefficient can be described as a function of both temperature (T) and atomic restoring force of the system (F(r)), especially around the defect region. The F(r) is a function of atomic distance, which depends on the temperature and defect distribution in the computational box. Therefore, when the temperature changes, the change of F(r) together with temperature results in non-monotone temperature dependence of the thermal expansion coefficient.

Figure 3b presents the effect of antisite defects on the thermal expansion coefficient of $UO_2$. Different from the effects of Frenkel defects, Figure 3b clearly indicates that for the concentrations of antisites investigated in the present work, the thermal expansion coefficient of the system increases with an increase in temperature. When the temperature is 600 K, the thermal expansion coefficient decreases with an increase in antisite defect concentration. When the temperature is 900 K or 1200 K, the thermal expansion coefficient has similar values for systems containing antisite defects less than 2%, but decreases around 25–30% when the antisite defect concentration is 5%. When temperature is 1500 K, the thermal expansion coefficient has the same value for systems containing 1% to 5% antisite defects, which is lower than that of the perfect system. Comparing the results shown in Figure 3a,b, it could be found that antisite defects have stronger effects than Frenkel defects on the thermal expansion coefficient of $UO_2$.

### 3.2. Elastic Modulus of UO$_2$

After the calculation of elastic constants, the bulk modulus, Young's modulus, and shear modulus of perfect UO$_2$ are initially calculated from three independent elastic constants ($C_{11}$, $C_{12}$, and $C_{44}$) using Equations (2), (5) and (6). Figure 4 depicts the dependence of the bulk modulus of perfect UO$_2$ on temperature calculations of systems. For comparison, the experimental results from Belle et al. [33], the MD derived bulk modulus from Basak et al. [23], and ab initio data calculated by Wang et al. [26] are also included in this figure. It is clear that the present study has similar results to those obtained by previous MD calculations and experiments but lower than those from the VASP calculation. Figure 4 also indicates that the bulk modulus of perfect UO$_2$ derived in this study decreases with an increase in temperature, which has been confirmed in the previous study by Dorado et al. [34].

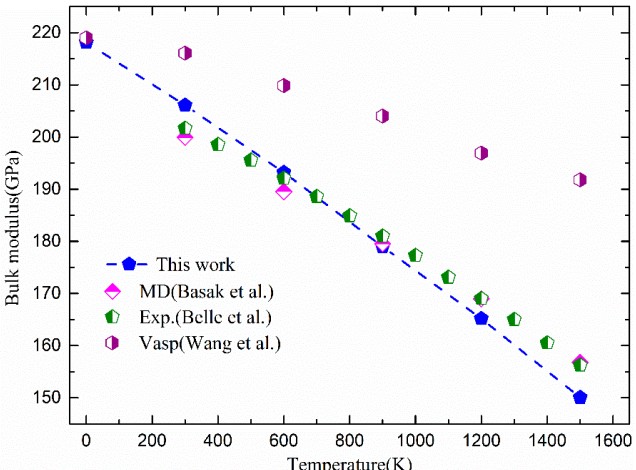

**Figure 4.** Variation of the bulk modulus of pure UO$_2$ versus temperature. For comparison, the experiment studies by Belle et al. [33], the Vasp data by Wang et al. [26] and the MD results by Basak et al. [23] are included.

Figure 5 shows the bulk modulus of the damaged UO$_2$ at different temperatures as a function of Frenkel defect concentration (dash) and antisite defect concentration (solid). Figure 5 shows that both a Frenkel defect and an antisite defect could considerably decrease the bulk modulus of UO$_2$, showing a linear decreasing dependence on temperature from 600 to 1500 K. With an increase in defect concentration, the elastic modulus also decreases accordingly, as shown by the figure. The extent of reduction in the elastic modulus for the system containing defects becomes smaller with increasing temperature and defect concentration. In addition, Figure 5 demonstrates that Frenkel defects increase the bulk modulus to a larger extent compared to that induced by antisite defects.

Figure 6 presents the effects of Frenkel (dash) and antisite defects (solid) on the shear modulus (*G*) of UO$_2$. Firstly, for the concentration of defects investigated in this work, the shear modulus decreases with an increase in temperature. The extent of reduction in the shear modulus decreases with increases in temperature from 600 to 1500 K. Similar to the effects on the bulk modulus shown in Figure 5, it can be seen from Figure 6 that the increase of defect concentration could significantly reduce the shear modulus of UO$_2$. However, the extent of reduction in *G* resulting from Frenkel defects is larger than that observed for antisite defects. For example, for 5% Frenkel and antisite defects there is a maximum of 20% and 17% reduction in *G* at all temperatures, respectively.

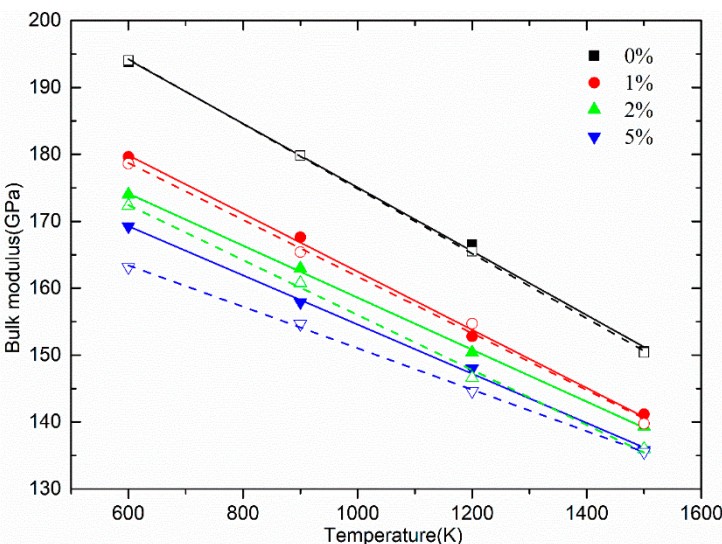

**Figure 5.** Variation of the bulk modulus of UO$_2$ containing different concentrations of Frenkel (dash) and antisite defects (solid) versus temperature. The fitted lines are also included in the figure.

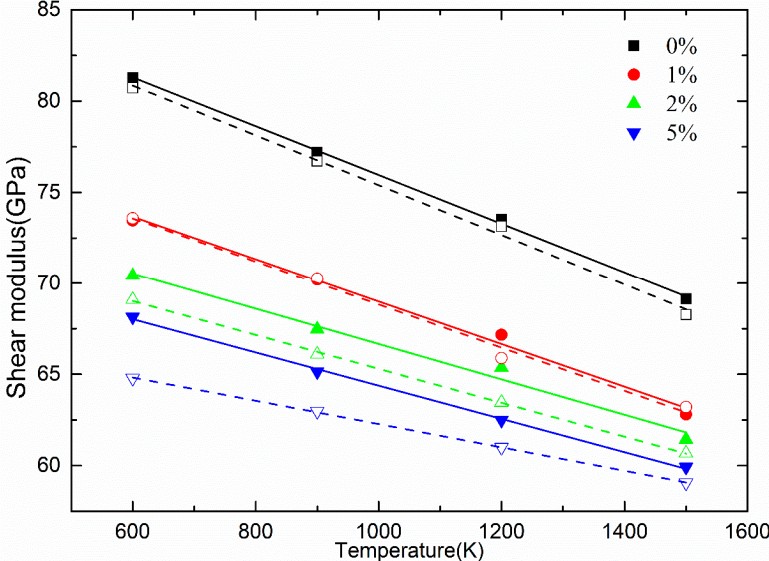

**Figure 6.** Variation of the shear modulus of UO$_2$ containing different concentrations of Frenkel (dash) and antisite (solid) defects versus temperature. The fitted lines are also included in the figure.

Young's modulus of UO$_2$ containing different concentrations of Frenkel (dash) and antisite defects (solid) as the function of the temperature is plotted in Figure 7. From this figure, it is clear that Young's modulus of uranium dioxide decreases linearly with the increase in temperature for Frenkel and antisite defects within the given concentrations. This result is similar to that reported by Jelea [13] et al. who observed that Young's modulus for damaged UO$_2$ with different percentages of porosity linearly decreases with increases in temperature. Similar to the results of the bulk modulus and the shear modulus, Young's modulus also decreases with increasing temperature and concentration of defects. The relative change in $Y$ due to Frenkel defects is larger than that observed for antisite defects. Within the given temperatures, a maximum of 20% reduction in $Y$ is observed for UO$_2$ systems containing 1%, 2%, and 5% Frenkel defects. In contrast, for antisite defects with the same concentrations, there is a maximum of 16% reduction in $Y$ at all temperatures.

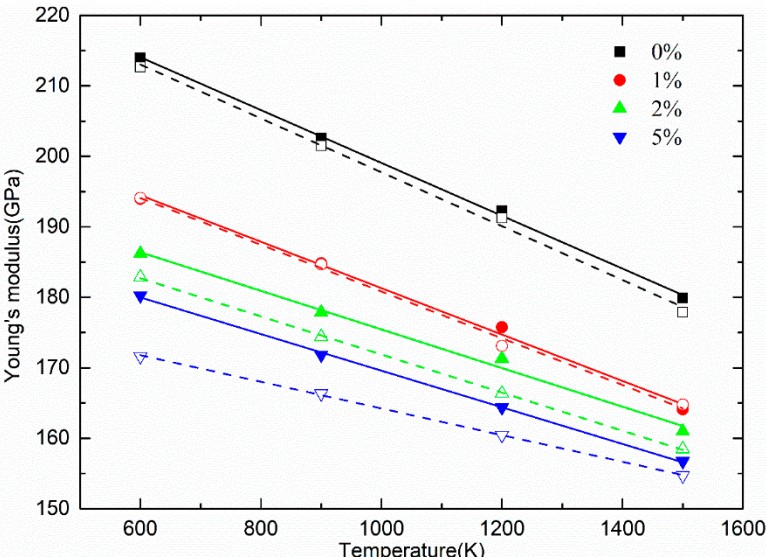

**Figure 7.** Variation of Young's modulus of UO$_2$ containing different concentrations of Frenkel (dash) and antisite (solid) defects versus temperature. The fitted lines are also included in the figure.

### 3.3. Reduction of Elastic Modulus of UO$_2$ by Frenkel Defects and Antisites

The effects of Frenkel defects and antisites on elastic modulus are reported in detail in Figures 8–10. In each of these figures, (a) and (b) represent the influence of Frenkel defects and antisite defects on the elastic modulus, respectively. The percentage of reduction ($R$) in the elastic modulus as the function of fractional point defects for all temperatures is plotted. The percentage of reduction of the elastic modulus is calculated using the following Equation (7).

$$R = (M_{\mathrm{p}} - M)/M_{\mathrm{p}} \tag{7}$$

where $M_{\mathrm{p}}$ and $M$ represent the elastic modulus of a perfect and defective UO$_2$, respectively.

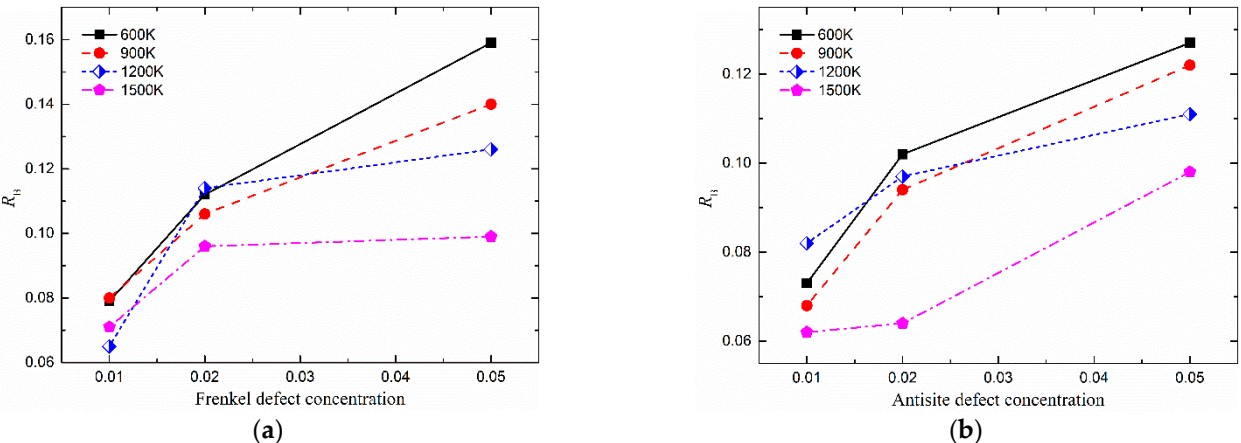

**Figure 8.** $R_{\mathrm{B}}$ as a function of the concentration of Frenkel defects (**a**) and antisites (**b**) at different temperatures.

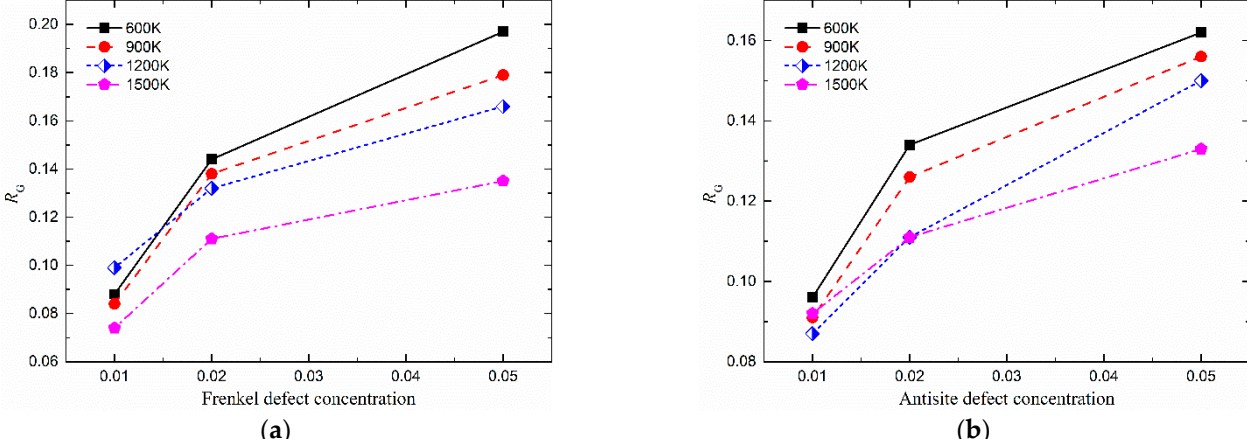

**Figure 9.** $R_G$ as a function of the concentration of Frenkel defects (**a**) and antisites (**b**) at different temperatures.

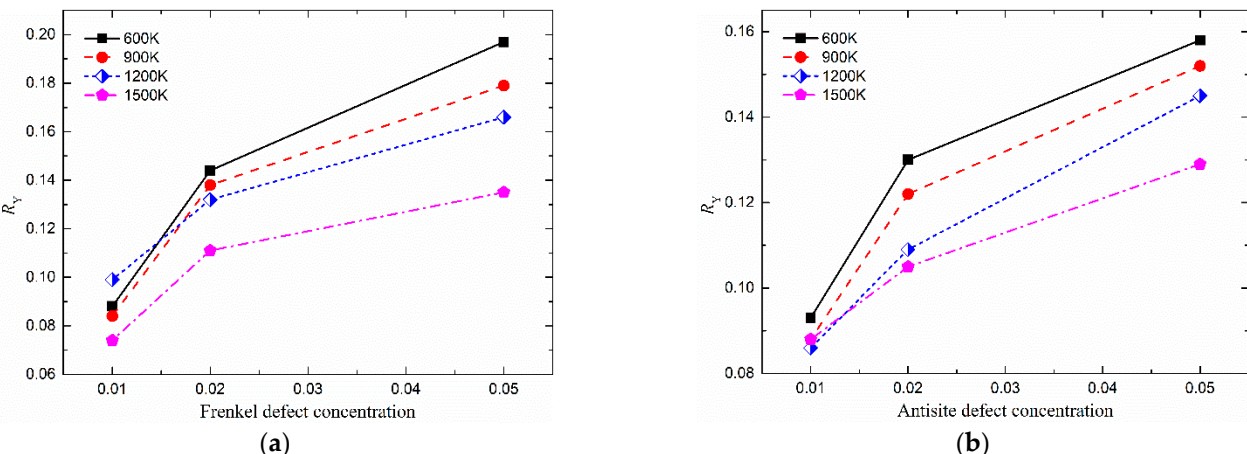

**Figure 10.** $R_Y$ as a function of the concentration of Frenkel defects (**a**) and antisites (**b**) for different temperatures.

Figures 8a, 9a and 10a illustrate that in low concentration ranges the difference among the percentages of reduction in the elastic modulus for different temperatures is small. However, this difference is more evident at higher defect concentrations. In addition, this is not as significant for antisite defects as it is for Frenkel defects. The *R* of the elastic modulus due to antisite defects is smaller than that by Frenkel defects. For 5% defects, the degradation of the bulk modulus, shear modulus, and Young's modulus by Frenkel defects is higher than that of antisite defects by 3.4, 3.9, and 2.9%. Therefore, by comparing Figures 8–10, it can be observed that the percentage of reduction in the elastic modulus by Frenkel and antisite defects follows the trend $R_f > R_a$ for all studied defect concentrations.

## 4. Conclusions

In order to characterize the effect of irradiation on the performance of the nuclear fuel, it is necessary to investigate how the irradiation defects affect the thermal-mechanical property of nuclear fuel. In this study, the impact of Frenkel defects and antisites on thermal expansion and elastic constants has been examined in $UO_2$ via the molecular dynamics method in the temperature range of 600 to 1500 K. The results indicate that both Frenkel defects and antisite defects reduce the thermal expansion coefficient. However, the reduction in the thermal expansion coefficient due to antisite defects is larger than that observed for Frenkel defects. For the elastic modulus, the calculated bulk, shear, and Young's modulus of the pure $UO_2$ are in agreement with the experimental values.

Furthermore, the present results indicate that both Frenkel defects and antisite defects reduce the elastic modulus at all temperatures. The degree of reduction in the elastic modulus increases with increasing concentrations of defect. In addition, the percentage of reduction in the elastic modulus due to Frenkel and antisite defects follows the trend $R_f > R_a$ at all studied defect concentrations. All these calculated values can be used to predict the performance of $UO_2$ under irradiation used in the nuclear reactor environment.

**Author Contributions:** Conceptualization, Z.W. and N.G.; methodology, Z.W.; software, M.Y.; validation, C.Y., X.L. and Z.W.; formal analysis, Z.W.; investigation, M.Y.; resources, N.G.; data curation, Z.W.; writing—original draft preparation, Z.W.; writing—review and editing, N.G.; visualization, M.Y. and L.D.; supervision, N.G. and Z.Y.; project administration, Z.Y.; funding acquisition, N.G. and X.W. All authors have read and agreed to the published version of the manuscript.

**Funding:** This work was financially supported by the National Natural Science Foundation of China (Project Nos. 12075141 and 12175125).

**Institutional Review Board Statement:** Not applicable.

**Informed Consent Statement:** Not applicable.

**Data Availability Statement:** The data that support the findings of this study are available from the corresponding author upon reasonable request.

**Conflicts of Interest:** The authors declare no conflict of interest.

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
