# Peer review of "Effect of Radiation Defects on Thermo–Mechanical Properties of UO2 Investigated by Molecular Dynamics Method"

_metals, doi:10.3390/met12050761_

Round 1

Reviewer 1 Report

The authors study the effect of two kinds of defects in a crystalline structure of UO2 using molecular dynamics. They report a dependence of the equilibrium lattice parameter, thermal expansion coefficient and elastic moduli on temperature. Their results for a perfect UO2 crystal are compared to available experimental or theoretical data and exhibit a good agreement with most of the already published data. Their results for a system containing irradiation induced defects such as the Frenkel and antisite defects are the new information contained in the present manuscript. The text is written relatively clearly with just minor language errors.

Considering the agreement of the presented data with literature, I believe that the results are correct and trustworthy, however, the authors should pay attention to the following issues:

I would like to ask the authors to check what is the uncertainty of their results and mention it in the discussion. Particularly in Fig. 3a one isn’t sure if the effect of 1% concentration is really opposite to that of 2% and 5% concentrations or is almost negligible (and the values are practically same as for zero concentration within the confidence interval).

Have the authors check the concentration of defects after the equilibrating process? Are they sure that, e.g., the Frenkel defects did not annihilate at least partially at elevated temperatures?

Reviewer 2 Report

Uranium dioxide is one of the widely used nuclear fuels in world nuclear industry. Different radiation defects (e.g., vacancies, voids, interstitials and so on) resulting in the severe degradation of the physical, thermal and mechanical properties of  nuclear fuels may carry a potential hazard to the operation of nuclear reactors. Therefore, this study has a practical relevance.

The comments on the paper are as follows:

The authors rely on the new LAMMPS methodology. This is a new methodology (reference of 2022), so a reader of Metals needs more details of the authors’s calculation to be presented. Along the article, the authors repeatedly compare their results with the work of other authors using the MD approach. Therefore, it is very useful to give the main ideas of LAMMPS and how they differ from the MD.

The authors consider in the paper only the Frenkel defects and antisites. But the question remains: can the Schottky defects occur under the conditions they consider? A detailed answer why the authors do not consider the Schottky defects should be done.

The article is characterized by the complete lack of any interpretation of the simulation results. Interpretation is very important especially where the results obtained are not compared with experiment. For example, thermal expansion coefficient presented on Fig.3. How can the non-monotone temperature dependence of the thermal expansion coefficient be explained? The saturation effect of dependence of UO2 lattice constant on defect concentration is also unclear (Fig.2). I insist on the authors to give some qualitative interpretation of the results obtained.

Reviewer 3 Report

The authors' research on the effects of radiation defects on the thermo-mechanical properties of UO2 is timely and interesting and should be published in Metals subject to the following minor amendments:

(1) line 22: "UO2" subscript 2

(2) line 64: "UO2" subscript 2 (and lines 72, 79, 102, 168, 170, 183, 202, 235)

(3) line 80: "CaF2" subscript 2

(4) Section 2: The authors should state explicitly whether the simulations have been checked for convergence and whether thus the final results are robust.

(5) Figure 1: The present results are rather difficult to discern in this figure. Perhaps the orange squares should be enlarged or brought into the foreground.

(6) Line 142: Can the authors provide a possible explanation for the difference in this behaviour?

(7) Figure 3 and line 216: "defected" should probably read "defective"

(8) Line 171: "C11, C12 and C44" subscripts

(9) Line 262: Remove LaTeX symbols

(10) References 9, 10, 24, 25, 27, 30: Name of the journal missing

(11) Reference 32 seems incomplete
